# Organ Specific Copy Number Variations in Visceral Metastases of Human Melanoma

**DOI:** 10.3390/cancers13235984

**Published:** 2021-11-28

**Authors:** Orsolya Papp, Viktória Doma, Jeovanis Gil, György Markó-Varga, Sarolta Kárpáti, József Tímár, Laura Vízkeleti

**Affiliations:** 12nd Department of Pathology, Semmelweis University, 1091 Budapest, Hungary; orsolya.papp@turbine.ai (O.P.); domaviki@gmail.com (V.D.); vizkeleti.laura@med.semmelweis-univ.hu (L.V.); 2Turbine Simulated Cell Technologies, 1027 Budapest, Hungary; 3Department of Dermatology, Venerology and Dermato-Oncology, Semmelweis University, 1085 Budapest, Hungary; karpati.sarolta@med.semmelweis-univ.hu; 4Division of Oncology, Department of Clinical Sciences, Lund University, 221 84 Lund, Sweden; jeovanis.gil_valdes@med.lu.se; 5Clinical Protein Science & Imaging, Department of Biomedical Engineering, Lund University, 221 84 Lund, Sweden; gyorgy.marko-varga@bme.lth.se; 6Chemical Genomics Global Research Lab, Department of Biotechnology, College of Life Science and Biotechnology, Yonsei University, Seoul 03722, Korea; 71st Department of Surgery, Tokyo Medical University, Tokyo 160-8582, Japan

**Keywords:** distant organ metastasis, DDR deficiency, HGF/MET autocrine activation, immunogenic mimicry, BRAF and NRAS mutant allele frequency

## Abstract

**Simple Summary:**

Malignant melanoma is a highly metastatic disease disseminating to several distant sites. This potential is also of great clinical impact for patient survival and therapeutic success. Knowledge about melanoma genomics is mainly based on lymphatic or skin metastases derived data, whereas data from distant sites is limited. Therefore, an autopsy-based visceral metastasis biobank was established, and an array-based copy number variation (CNV) analysis was performed, focusing primarily on major organs (brain, lung, and liver) and completed partly by proteomic analysis. A unique picture emerged about organ-specific CNV-type distributions or gene alterations, including the frequent loss of DNA damage error genes in brain metastases, the presence of HGF/MET autocrine loop in brain and lung metastases, the traces of immunogenic mimicry exclusive for lung metastases or the correlation of *BRAF* copy number and mutant allele frequency, especially in lung metastases. All these above phenomena have a great influence on therapy efficacy or resistance.

**Abstract:**

Malignant melanoma is one of the most aggressive skin cancers with high potential of visceral dissemination. Since the information about melanoma genomics is mainly based on primary tumors and lymphatic or skin metastases, an autopsy-based visceral metastasis biobank was established. We used copy number variation arrays (*N* = 38 samples) to reveal organ specific alterations. Results were partly completed by proteomic analysis. A significant increase of high-copy number gains was found in an organ-specific manner, whereas copy number losses were predominant in brain metastases, including the loss of numerous DNA damage response genes. Amplification of many immune genes was also observed, several of them are novel in melanoma, suggesting that their ectopic expression is possibly underestimated. This “immunogenic mimicry” was exclusive for lung metastasis. We also provided evidence for the possible autocrine activation of c-MET, especially in brain and lung metastases. Furthermore, frequent loss of 9p21 locus in brain metastases may predict higher metastatic potential to this organ. Finally, a significant correlation was observed between *BRAF* gene copy number and mutant allele frequency, mainly in lung metastases. All of these events may influence therapy efficacy in an organ specific manner, which knowledge may help in alleviating difficulties caused by resistance.

## 1. Introduction

Skin melanoma is among the most metastatic human cancers and is also among those with the highest tumor mutation burden (TMB~10^3^) due to UV-induced carcinogenesis [1]. Furthermore, chromosomal instability (CIN) results in a high frequency of copy number variations (CNV) in a similar range as TMB as determined by the highest resolution technologies such as whole genome sequencing (WGS) or whole exome sequencing (WES) [2]. Meanwhile the TMB and CNV of melanoma are not strictly tightened together [3]. Mutational profile of skin melanoma is well characterized as well as those of CNVs, the latest of which can be used to differentiate between premalignant and malignant pigment cell neoplasia: Loss of heterozygosity (LOH) of CDKN2A (9p21.3) and MYB (6q23.3) as well as amplification of CCND1 (11q13.3) and RREB1 (6p24.3) can be routinely used in pathological diagnostics [4]. Furthermore, skin melanoma is characterized by frequent amplifications of BRAF (7q34), NRAS (1p13.2), KIT (4q11) oncogenes (major mutated genes in melanoma) but also by EGFR (7p11.2) and MET (7q31.2) genes [5,6,7,8]. Conversely, LOH frequently affects PTEN (10q23.3) in melanoma [5,7,8]. Copy number alteration (CNA) analysis of primary melanomas identified 77 minimal homozygous deletion signature, which carried negative prognostic impact [9]. Another study demonstrated that whole genome duplications are responsible for the majority of copy number gains (CNG) in melanoma, while LOH is responsible for the majority of copy number loss (CNL) [10]. 

Genomic analysis of circulating melanoma cells identified 249 CNAs consisting of 139 CNGs and 110 CNLs of >50% frequency [11]. Further analysis of the top 37 CNAs defined 5-CNG signature (CNG:1p35.1, 2q14.3, 14q32.33 and CNL:14q32.11,21q22.3) with prognostic impact [11]. Other studies identified individual CNVs during metastatic progression: amplification of MITF (3p13) or NEDD9 (6p24.2) metastasis genes, as well as CDK6 (7q21.2), BRIC5 (17q25.3) and TEAD (11p15.3) [5,7]. It is also characteristic during metastatic progression that PTEN or the metastasis suppressor, KISS1R (19p13.3) are lost [7]. However, it is not well known if these genetic alterations during progression are uniform or progression-type specific. 

Melanoma characteristically produces distant skin metastases as part of their homing process or distant lymphatic metastases, but clinically, the most relevant metastases are the visceral ones involving major organs such as brain, lung or liver, although other organs can also be affected. Our knowledge on the genomics of skin melanoma progression is mostly derived from biobanks dominated by lymphatic or skin metastases, and data from visceral metastatic melanoma are much more limited. An analysis of melanoma brain metastases revealed frequent deletion of CDKN2A/B locus, which had a negative prognostic impact [12]. A more recent analysis of melanoma brain metastases detected high frequency of EGFR amplification in metastases overall but the highest in the brain [13]. This study, although at lower frequency, discovered MET amplification in melanoma brain metastases as well. 

We have established an autopsy-based melanoma metastasis biobank [14], which was used to analyze CNV alterations during the visceral progression of skin melanoma. 

## 2. Results

### 2.1. Metastatic Melanoma Cohort

We collected an autopsy-based fresh frozen melanoma metastasis biobank of ten cases containing 28 samples of brain, liver or lung metastases, where the primary tumors were formalin-fixed, paraffin-embedded (FFPE) materials of the resected tumors (Table 1, Appendix A). Histologically, the primary tumors were classical skin melanomas, and the cohort does not contain rare variants. The duration of the disease was on average about 6 years. Sixty percent of this small cohort was BRAF mutant, 30% was NRAS mutant and only one case was triple negative (KIT wild type too). The vast majority of the patients were treated with IFN2a, three patients were treated with BRAF inhibitor (BRAFi) while 2–2 patients each have been treated either with Dacarbazine (DTIC) or platina-based chemotherapy. No patients were treated with immune-oncology drugs. 

### 2.2. CNV Landscape of Primary Tumors and Metastases of Skin Melanoma

The CNV landscape was identified in 38 primary melanomas and metastases using OncoScan FFPE and CytoScan HD arrays (Figure 1). Only structural CINs were examined since whole genome duplication was not observed in any of the investigated samples. Compared to primary tumors, distant organ metastases showed frequent (>35%) gains of several large chromosomal regions at chromosome 1, chromosomal arms 5p, 9q, 14q, 16p, 17q, 22q and losses at 6q, 9p, chromosome 10, 11q. Conversely, gains at 4p and 10q were characteristic to primary melanomas exclusively. Meanwhile, common aberrations shared by the primary and metastases were also represented across the entire genome, especially on chromosome 7, 8, 20 and chromosomal arm 6p (Figure 1). Regarding the comparison of distant melanoma metastatic sites, liver and lung metastases showed a more similar CNA pattern to each other than brain metastases. 

Genomic instability increased significantly (*p* = 0.045) parallel to the tumor progression, when alterations of primary melanomas were compared to the pooled metastases or individual metastases, except the one in the liver (Figure 2, Appendix A). A more detailed analysis of genomic alteration types provided a refined picture of visceral progression of melanoma. Most of the genomic alterations fall into the low-copy number gain (lCNG) category, but this was not significantly different in metastases. Conversely, high-copy number gain (hCNG) as well as all the three-copy number loss (CNL) types (homozygous, heterozygous and LOH) significantly increased in visceral metastases as compared to primary tumors with significant differences in individual metastases. High-CNG was the lowest in brain metastases and the highest in lungs. In contrast, lCNG and heterozygous-CNL (heCNL) was the highest in brain metastases and the lowest in liver metastases, similar to homozygous-CNL (hoCNL). Copy-neutral LOH (cnLOH) was also observed to be significantly increased in metastases, and it was less frequent in brain metastases and the most persistent was in liver.

Since the average percentage of genomic alterations was higher in metastases, we tried to examine what kind of governing features could be hindered behind this phenomenon. We observed frequent heCNL events in regions bearing genes important in the DNA damage repair (DDR) processes. Therefore, we examined the CNA status of 43 genes in four DDR pathways involved in both single- and double-strand DNA error repairs, namely base excision repair (BER), nucleotide excision repair (NER), mismatch repair (MMR) and homologous recombination (HR). Overall, metastatic melanomas were enriched in heCNLs affecting DDR genes, especially brain metastases, where we detected the highest number of alterations of genes acting in HR and NER (Figure 3). The most predominant genes were *APTX*, *FEN1*, *GTF2H5*, *DNA2*, *MUS81* and *TOP3A* with varying incidences between visceral sites. 

### 2.3. Marked Genetic Differences between Visceral Metastases of Melanoma

Searching for organ specific CNV alterations, we primarily focused on the presence of hCNGs (CN ≥ 4) and hoCNLs, as cutaneous melanoma usually shows a high genetic instability increasing with progression (Figure 4). 

Overall, lung metastases had the highest number of alterations. All of these were hCNGs and affected 25 chromosomal regions including a total of 3010 genes. Out of these genes, 1440 genes (47.8%) on 10 loci (5q34-p35, 6p12, 7q11, 7q21-22, 8p21, 8q22, 19p13, 20p11-p13, 20q11, 22q11-q13) were exclusive for melanoma lung metastases (Figure 4, Appendix A). Searching for alterations associated with metastatic ability, lung metastases were characterized by *NEDD9*, *TEAD1/2/4*, *SNAI1* and *TWIST1/2* amplifications. Lung metastases uniquely contained the amplification of *CDK6*, *MAPK1* and *ABL2* (Table 2). Chromosomal regions, including 19 immune cell (dendritic cells, T cells, B cells, macrophages) specific genes, were frequently harbored by amplification in lung metastases and partially in liver ones (Table 3), suggesting that immunoselection might play a significant role exclusively at this site. 

Regarding liver metastases, 19 regions with 1560 genes exhibited amplification. Only a low proportion (25 genes, 1.6%) of genes located on two loci (6p21, 8p23) was unique to liver and more than 98% of genes (N = 1530) was overlapped with ones found in lung metastases (Figure 4, Appendix A). Searching for historical metastasis genes, amplification of *TWIST1* and *SNAI2* together with *S100A9/10/11/12* and *MAPK15* kinase was observed (Table 2). Regarding the previously mentioned immunogenic pattern of lung metastases, only *CD40/83/172* gene amplification was found to be a shared feature between lung and liver metastases (Table 3).

Melanoma brain metastases showed a different picture. High-copy gain was observed in only four chromosomal regions including 299 genes. A total of 55.5% of these genes (N = 166) on 3 loci (1q21, 5p15, 8q24) was only characteristic for the examined brain metastases (Figure 4, Appendix A). We found *TERT* and *CD160* amplification as a unique hCNG of brain metastasis (Table 2 and Table 3). Homozygous loss of 9p21.3 locus (28 genes) was exclusive for this distant site, including several interferon genes as well as *CDKN2A* and *B*.

We also analyzed the distribution of driver oncogene-specific CNA types per distant metastatic sites and independently from visceral localization as well. The proportion of LOH was higher in *NRAS* mutant samples in case of liver metastases. Furthermore, a marginal statistical difference was found in the incidence of hoCNL in both brain and liver samples in favor of *NRAS* oncogene mutation (Appendix A). Independently from the metastatic site, homozygous loss of 1q24 and 15q24 regions was observed in *NRAS* positive samples only, while hCNGs could be observed across the entire genome. Shared alterations were observed on 1q, 3q, 5q, 7p, 9q, 13q, 14q and 16p (Appendix A).

HGF/MET autocrine activation is a well-known process in the development of malignant melanoma. The co-occurrence of copy number gain of MET receptor and its ligand, HGF was the most pronounced in brain metastases (5 out of 10 examined cases). By correlating the probe medians of MET and HGF genes, a strong significant correlation was found in lung metastases (Pearson’s Rho; R = 0.699, *p* = 0.036; Appendix A).

### 2.4. Expression Profiles of the Immunogenic Mimicry Related Genes

The expression profiles of the 19 genes identified with higher copy numbers in metastatic melanomas were explored in three different validation cohorts at the *ProteoGenomic*; transcript and/or protein level. The transcript data from 443 melanomas were curated and processed from the Cancer Genome Atlas (TCGA) repositories [15]. The clinical protein expression profiles were generated from a prospective, and a postmortem study cohort, recently presented within the Human Melanoma Proteome Atlas [16,17]. In summary, only one out of 19 amplified immunogenic mimicry (IGM) genes (CD172) was not detected at transcript levels in TCGA, while at the protein level, 10 proteins were identified in each proteomics cohort. Herein, 9 out of the 19 genes were quantified in the three datasets (Table 3, Appendix A). The transcriptomics cohort was divided according to the sample origin in: (1) primary tumors, (2) cutaneous, (3) lymph node, and (4) distant metastases. The comparison within the expression levels showed significant differences in 13 genes between the primary and at least one group of the metastases. With the exception of CD1A, all presented higher levels in the metastases (Figure 5). This is particularly true for the lymph node metastases. 

Conversely, at the proteome level, seven proteins indicated significant differences. In the prospective cohort, two proteins; CD93 and CD84, displayed higher levels within the tumor samples when compared to non-tumors (Figure 6A). In the case of CD247, it was not detected in non-tumors and additionally was found to be upregulated in metastases from the lymph node group, when compared to primary tumors. In addition, two proteins; CD1A and CD70 were downregulated during the progression of the disease. We also observed within the study that only two proteins showed significant differences between different groups of metastases from the postmortem cohort (Figure 6B). This was particularly clear for CD320, that was differentially expressed between the lung and liver metastases groups. Altogether, the expression profiles of these genes indicate to a large extent, that an upregulation of these genes occurs in metastatic melanomas, as compared to the primary tumors.

We also observed that there is an upregulation within the expression profiles of melanomas in general when compared non-tumor samples at the protein level. However, the differences between metastases originating in different locations were found to befall at a lower level.

### 2.5. Mutant Allele Frequency (MAF) Changes of Driver Oncogenes during Progression

Since we observed frequent CNG events on large chromosomal regions of chromosomes 1 and 7, where *NRAS* and *BRAF* genes are localized, we compared copy number alterations of these two genes to the changes of mutant allele frequency. Comparing to the matched primary tumor, proportion of *NRAS* mutant clones was unchanged or higher in metastatic samples (Table 4). All the examined samples were diploid for *NRAS* gene. However, a 120,000 kbp long cnLOH region, including the SNP variant of *NRAS* gene at 1p13.2 locus as well was observed in case of brain metastasis of the MF53 sample (*NRAS^Q61L^*) and primary tumor of the No. 55 sample (*NRAS^Q61R^*). In parallel to this, MAF was more than three times higher in the No. 53 brain metastasis, whereas no change of MAF was noticed in the case of No. 55 sample pairs. 

*BRAF* gene was usually observed in distant metastatic melanomas with increased *BRAF* MAF value comparing to the primary tumor [14]. *BRAF* high copy gain was the most typical for lung metastases, often bearing an average of 4-5 copies per cell (Table 4). A moderate, but significant correlation (R = 0.540, *p* = 0.009) was observed between *BRAF* gene copy number alteration and mutant allele frequency in *BRAF* mutant metastases compared to their matched primary tumors (Table 5). We determined the threshold of mutational homozygosity from MAF ≥ 60%. Only five samples met these criteria, three of which we also found bearing LOH, which can hypothetically boost the development of homozygosity.

## 3. Discussion

Chromosomal instability (CIN) is one the principal forms of genomic instability, which is known to be a hallmark of cancer. CIN as a definition refers to enhanced rate of chromosomal mis-segregation, which is one of the main driving forces of the development of aneuploidy. As every other alteration in the genome of cancer cells, CIN has the bidirectional power to both increase or decrease cell survival. Gaining additional copies of oncogenes can be advantageous to the cell, just as homozygous loss of essential tumor suppressor genes. It is a long-known phenomenon, that CIN is an enduring process occurring already in primary tumors in the early phase of tumor progression. We compared and analyzed the frequent (>35%) copy number gains (CNG) and losses (CNL) captured in the genome of melanoma patients with visceral metastases. We also found that genomic instability increased in both pooled metastases and individual ones except for liver when the copy number variation (CNV) landscape of primary tumors was compared to the CNV pattern of visceral metastases. When we delved into these frequent CNVs, we generally observed that most of the alterations were low CNGs. We also explored a statistically significant increase of high CNGs in metastases in an organ specific manner, the number of these kind of aberrations was the highest in lung metastases and the lowest in brains, while low CNG and both heterogenous and homogenous CNLs were predominant in brain metastases.

Homologous recombination deficiency (HRD) characterizes a unique set of cancers of various histologic origin leading to increased tumor mutation burden and special sensitivity to immune checkpoint inhibitors or to PARP inhibitors [18]. A study demonstrated that in breast cancer disseminated brain metastases genomic aberration-based HRD measures are significantly higher in metastases relative to their matched primary tumors suggesting that HR-deficient brain metastases might be more sensitive to PARP inhibitor treatment [19]. Another recent analysis of human melanoma found that 21.4% of this tumor type has at least one HRD gene mutation including *BRCA1*, *ARID1A*, *ATM*, *ATR* and *FANCA* [20]. To see whether there is a possible connection between increasing rate of genomic errors and DNA damage repair (DDR) defects regarding melanoma, we examined the CNA status of more than 40 genes whose protein products act in this pathway module. Herein, we found a common occurrence of heCNL events harboring genes important in the repair machinery in visceral metastases of human melanomas compared to primary tumors. In melanoma brain metastases, heterozygous copy number loss of HR and SSB repair, but no mismatch repair-associated genes, was found, which from the most prevalent genes were *APTX*, *FEN1*, *GTF2H5*, *DNA2*, *MUS81* and *TOP3A*. Accordingly, the loss of DNA repair function could be underestimated, at least in melanoma.

The recent science milestone on “The Human Proteome Project High-Stringency Blueprint”, has opened up new avenues on the ability to align the gene-protein functional relationship at a cellular level that previously was unprecedented [21]. We have observed the amplification of a wide variety of immune genes in metastatic melanoma. However, out of these 19 genes, five were not found at protein levels in validation sets (*CD37*, *CD83*, *CD160*, *CD172*, *CD244*). Expression of immune cell genes by various cancers is a well-known phenomenon, where the best-known example is the expression of PD-L1 [22,23]. The overrepresentation of PD-L1 in tumor cells is rarely due to gene amplification, rather due to epigenetic mechanisms. [24]. There are several other examples for the ectopic expression of immune cell genes in cancer, such as *CD36*, *CD70*, *CD40*, *CD47*, *CD172* and *IDO1* [25,26,27,28,29]. Most of these studies reported these ectopic expressions as part of the immune escape of cancers. Except CD70, all these genes have been reported in human melanoma as well. Furthermore, *CD36* and *CD83* expressions were detected in human melanoma cells, but we were not able to confirm it in our studies [30,31]. Meanwhile, our observation of gene amplification and expression of *CD1A*, *CD48*, *CD84*, *CD93*, *CD209*, *CD247*, *CD320*, *IDO2, BCR* and *IL17R* in human melanoma is novel. It is also intriguing to see that this immunogenic mimicry characterizes the lung metastases exclusively, suggesting that this type of immunoselection may takes place exclusively in lung metastasis and not in liver or brain metastasis. Liver is an organ of the reticuloendothelial system (RES) where the innate immune system can be functional to be involved in immunologic selection for metastatic cells, while the brain is a unique immunological microenvironment where the function of antitumoral immunity is not well described. 

There is another aspect of this immunogenic genotype (at least in melanoma) which has to be considered. Characterization of tumor microenvironment especially immune cells is mostly based now on bioinformatic analysis of RNA-seq-expression data [26,32] but not immunohistochemistry (IHC). Such an analysis is based on the presumption that immune cell markers are solely expressed by immune cells and does not consider the possibility of ectopic expression by cancer cells (in our case by melanoma cells). Such an analysis may provide false data on the exact composition of immune cells infiltrate. At least in human melanoma these analyses must be double checked by IHC analysis to exclude the possibility of interpreting false data. 

Contrary to melanocytes, it is a well-known fact that melanoma cells have the ability to express c-MET and also to release HGF, thus activating c-MET in an autocrine fashion [33]. Stimulation of the HGF/MET pathway strengthens numerous processes that are essential for melanoma development and construction of visceral metastatic niche [34]. Autocrine activation of MET by HGF is reported to be an influencing factor in immediate resistance to RAF inhibitors in BRAF positive melanomas as a bypass mechanism by reactivating the mitogen-triggered protein kinase (MAPK) and phosphatidylinositol-3-OH kinase (PI(3)K)-AKT signaling pathways [35]. Mutation of *MET* was considered to be a melanoma progression marker [13]. A recent large-scale analysis of mutational networks in cancer revealed 23 distinct such driver gene networks two of which were assigned to malignant melanoma [36]. One of these mutational networks presented in melanoma was the one containing *MET* besides *CARD11*, *CBLB*, *PPP6C* and *RAC1*. Here, we provide supporting evidence for the frequent co-occurrence of copy number gain events affecting the MET receptor and its ligand, HGF especially in brain metastases. In addition, array CGH derived probe medians of these two genes showed strong significant correlation in case of lung metastases in our melanoma cohort. The same co-amplification of *MET* and *HGF* is reported to be a relatively frequent genetic aberration of various cancer types including lung cancer or renal cancer serving as a useful target for anti-MET therapies [37].

Melanoma is a tumor that is able to disseminate their tumor cells relatively early during cancer progression. It is now a well-known fact that melanoma cells have multiple forms of movement during migration, including collective and individual forms [38]. The most studied forms of individual migration are the elongated-mesenchymal and rounded-amoeboid mode which between melanoma cells can easily shift in order to migrate tumor cells effectively towards a new prestructured recipient microenvironment [39]. The hub regulators of the bidirectional transition between these motility modes are Rac and Rho GTPases, whose activity are modified by numerous other interactors also included in our dataset as amplified genes, such as *NEDD9*, *TWIST*s, *SNAI*s, *TEAD*s and *ABL* [7,40,41,42]. Beside the aforementioned motility associated genes, high-copy number gain of cell proliferation regulators (*CDK6*, *MAPK1*, *MAPK15*) were also observed in both lung and liver metastases, while in brains the amplification of *TERT* and deletion of *CDKN2* genes were represented only. According to Rákosy et al. clear homozygous deletion of the 9p21 locus bearing *CDKN2* is usually infrequent in primary cutaneous melanomas [43]. Therefore, its relatively frequent presence in brain metastases may predict higher metastatic potential to this distant site. These results suggest lung and liver metastases, regarding the pathway membership of the coded proteins, were more similar to each other than to brain metastases.

Approximately 50% of cutaneous melanoma patients are mutant in *BRAF* gene, 15–20% of patients are positive for *NRAS* damaging mutations both leading to a constitutively enhanced MAPK pathway activity [44]. The efficacy of targeted therapies such as BRAFi or MEK inhibitors are known to be weakened by an early evolving acquired resistance in melanoma patients. The amplification of *BRAF* and *NRAS* genes supporting the reactivation of MAPK pathway is one of the main reasons behind this phenomenon. A multicenter analysis focusing on acquired BRAFi resistance reported that 13% of the examined patient cohort had increasing *BRAF* gene copy number during tumor progression compared to the CNA status of the gene at the time of diagnosis [45]. The same effect was reported by several other studies about metastatic melanoma patients who did not respond to the targeted therapy due to *BRAF* amplification [46,47,48]. We detected frequent and common aberrations shared by the primary tumors and melanoma metastases on chromosome 7, where *BRAF* localizes, while CNGs on chromosome 1 bearing *NRAS* was a metastases-only feature. Because of this, we analyzed the potential connection between copy number alterations and mutant allele frequency (MAF) of *BRAF* and *NRAS* genes. Comparing to the matched primary tumor, proportion of *NRAS* mutant clones was unchanged or higher in metastatic samples but there was no detected copy number gain of *NRAS* gene. Copy number of *BRAF* gene was significantly correlating with increasing *BRAF* MAF values in visceral melanoma metastases compared to the matching primary tumors, especially in case of lung metastases [14]. It is a long-known fact, both about primary and metastatic melanomas, that chromosome 7 and chromosome 1 are frequently enriched by whole chromosome or chromosome arm affecting CNAs, which could contribute to the variations of mutant allele frequency in melanoma and therefore interfere with the clinical efficacy of mutant–protein targeted therapy [45,46,47,48].

## 4. Materials and Methods

### 4.1. Primary Tumors and Visceral Metastases

Frozen autopsy melanoma tissues were obtained from the 2nd Department of Pathology and the 1st Department of Pathology and Experimental Cancer Research at the Semmelweis University (Budapest, Hungary) within 48 h after death. FFPE (formalin-fixed paraffin-archived) samples were derived from several locations including the Semmelweis University (2nd Department of Pathology, 1st Department of Pathology and Experimental Cancer Research and Department of Dermatology, Venereology and Dermato-oncology), the Szent György University Teaching Hospital, Department of Dermatology (Székesfehérvár, Hungary), the Nyírő Gyula and Honvéd Hospitals (Budapest, Hungary), the National Institute of Oncology (Budapest, Hungary) and the National Institute of Clinical Neurosciences.

This study was approved by the Semmelweis University Regional and Institutional Committee of Science and Research Ethics (document no.: 191-4/2014) and was carried out according to all relevant regulations. Lesions were diagnosed and tumor content was determined on the basis of formalin-fixed paraffin-embedded tissue sections stained with hematoxylin–eosin. A total of 10 primary and 28 metastatic melanoma samples were used for the microarray analysis (Table 1).

### 4.2. DNA Extraction, Quality Control, and Microarray Hybridization

Tumor to normal ratio of the analyzed primary tumors was 76% (±20), while it was in the case of metastases 78.9% (±18). Genomic DNA was isolated using QIAamp DNA FFPE Tissue and QIAamp DNA Mini Kits (BioMarker Kft., Gödöllő, Budapest, Hungary) according to the manufacturer’s protocol. Quantity and quality of DNA were both established by NanoDrop ND-1000 UV–visible spectrophotometer (Thermo Fisher Scientific, Inc., Waltham MA, USA) and Qubit dsDNA HS Assay on a Qubit 1.0 fluorometer (Thermo Fisher Scientific, Inc., Waltham, MA, USA). DNA samples with 260/280 ratio more than 1.8 were used for further analysis. Fragment analysis of the previously isolated DNA was carried out applying 1% agarose gel electrophoresis as well as BioAnalyzer 2100 using High Sensitivity DNA Kit (Agilent Technologies, Inc., Santa Clara, CA, USA). Samples with appropriate quality were considered for labeling and hybridization to CytoScan HD or OncoScan FFPE microarrays (Affymetrix Inc., Santa Clara, CA, USA). Labeling, hybridization, and imaging setup were performed by UD GenoMed Medical Genomic Technologies Ltd. (Clinical Genomic Center, University of Debrecen, Debrecen, Hungary) according to the recent laboratory protocol using 600 ng of sample DNA. The data discussed in this publication have been deposited in NCBI’s Gene Expression Omnibus [49] and are accessible through GEO Series accession number GSE185165 (https://www.ncbi.nlm.nih.gov/geo/query/acc.cgi?acc=GSE185165, accessed on 30 September 2021).

### 4.3. aCGH Experiments and Analysis

Molecular inversion probes (MIP)-based arrays were used to identify genome-wide CNV in DNA extracted from FFPE tumor samples and its matched distant metastasis pairs. DNA provided from the manufacturer was used as controls in each array batch. Quality scores were calculated for each sample, and specimens with poor data quality were excluded from further analysis. CNVs were then segmented using the SNP-FASST2 (Fast Adaptive States Segmentation Technique) algorithm in ChAS (Affymetrix, Inc., Santa Clara, CA, USA). DNA samples with 120–200 bp fragment length were hybridized onto OncoScan FFPE arrays (N = 21), while less fragmented DNA samples, mainly from snap-frozen tissues were hybridized onto CytoScan HD arrays (*N* = 18). Visualization and statistical analysis to determine copy number alterations (CNAs) and LOH were performed using both Nexus Copy Number Discovery 8.0 (BioDiscovery, Inc., El Segundo, CA, USA) and ChAS 2.1.0.16 (r6634) (Affymetrix, Inc., Santa Clara, CA, USA) software working with the processed CHP files. We determined a significance threshold of 0.05, differential threshold of 25% and specified 1000 kbp as the maximum spacing between adjacent probes. To eliminate small copy number alterations, the minimum number of probes per segment was set at 25. To detect gains and losses, the following log2 ratio thresholds were set: ± 0.3 for gains and losses, 0.7 for high CN gains and −1.0 for homozygous deletions. In case of allelic events, LOH calls smaller than 2 Kbp and not overlapped by a minimum of 25 probes were removed from further analysis. To avoid sex bias, all probes on chromosomes X and Y were excluded. Altered regions of 100% overlap with segmental duplications were also excluded.

### 4.4. Driver Mutation Analysis and Defining Mutant Allele Frequency

Driver mutation status of cutaneous melanoma samples were determined during the routine diagnostic by both Sanger sequencing and NGS (pyrosequencing) as previously described (Doma et al. 2019). Pyrograms were then analyzed with the PyroMarkQ24 software (Qiagen, Hilden, Germany) to determine the proportion of mutant versus wild type allele. Determination of the allele ratio was based on the relative peak heights of the corresponding nucleotides. In all cases the obtained PyroMark MAF values were corrected for the tumor/normal cell ratio by multiplying PyroMark% by 100/x% tumor DNA [14].

### 4.5. Gene Expression Analyses of the IGM Related Genes

All the experiments were performed within the “Human Melanoma Proteome Atlas” project as described previously [16,17].

## 5. Conclusions

We conducted a comparative array genomic hybridization analysis of CNV patterns of primary melanomas and their matched visceral metastases, especially focusing on brain, liver and lung disseminations. We confirmed some well-known biomarkers already reported in melanoma progression, such as organ-specific alterations of historical metastasis genes and HGF-MET autocrine reactivation. We observed that in lung metastases the mutant allele frequency of BRAF is increasing parallel with BRAF copy numbers, which may predict acquired therapy resistance to targeted therapies at this site.

We also identified some novel biomarkers important in immune cell escape mechanisms detected in mainly lung metastases, and successfully validated the findings on proteotranscriptomic levels, suggesting that the lung may represent a unique site for the metastatic dissemination of human melanoma. Besides this, we also observed an enrichment of DDR affecting aberrations specific to brain metastases, which may attract the attention back to DDR targeting in cases of melanoma.

## Figures and Tables

**Figure 1 cancers-13-05984-f001:**
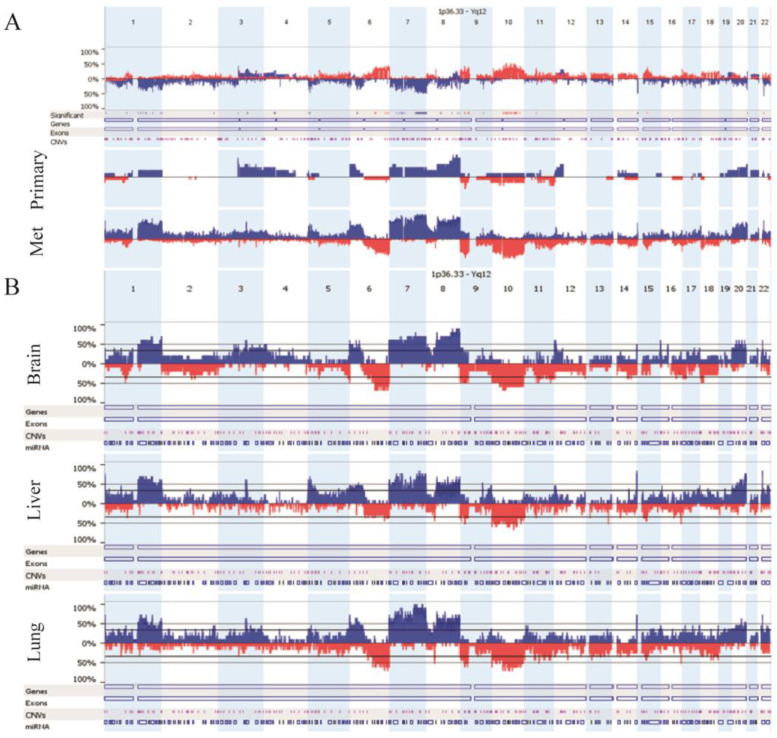
CNV landscape of examined melanoma samples: (**A**) primary tumors vs. distant melanoma metastases, (**B**) distinct distant metastatic sites (brain vs. liver vs. lung). Blue and red colors indicate copy number gains and losses, respectively.

**Figure 2 cancers-13-05984-f002:**
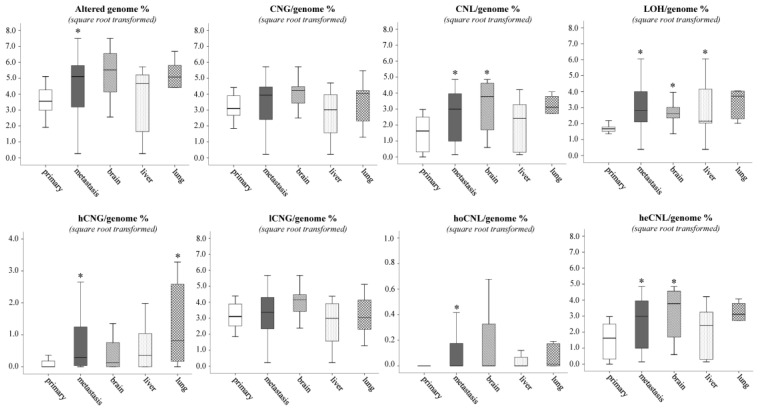
Distribution of CNV types between primary tumors and distant metastases from distinct sites. Asterisk means the level of significance (* *p* ≤ 0.05). Kruskal–Wallis test was used for the multiple group comparisons, and Mann–Whitney–Wilcoxon test was applied for the primary vs. all metastasis analysis. Abbreviations: CNG, copy number gain (CN > 2); CNL, copy number loss (CN < 2); LOH, loss of heterozygosity; hCNG, high-copy number gain (CN ≥ 4); lCNG, low-copy number gain (4 > CN > 2); hoCNL, homozygous copy number loss (CN = 0); heCNL, heterozygous copy number loss (2 > CN > 0).

**Figure 3 cancers-13-05984-f003:**
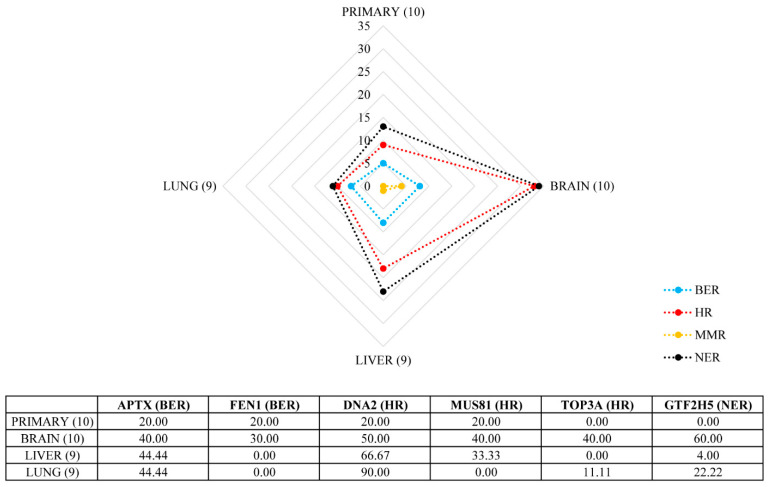
Copy number loss (CNL) frequency in different melanoma metastases affecting genes coding proteins in DDR subpathways. Radar chart represents the CNL frequency in primary melanomas and visceral metastases. In brackets, we indicated the count of samples resected from primary tumors and a given metastatic site. The vertical axis represents the number of genes altered in any of the DDR subpathways. Table shows the frequency (percentage) of the TOP6 DDR genes affected in melanoma. Abbreviations: DDR, DNA damage repair; BER, base excision repair; HR, homologous recombination; MMR, mismatch repair, NER, nucleotide excision repair.

**Figure 4 cancers-13-05984-f004:**
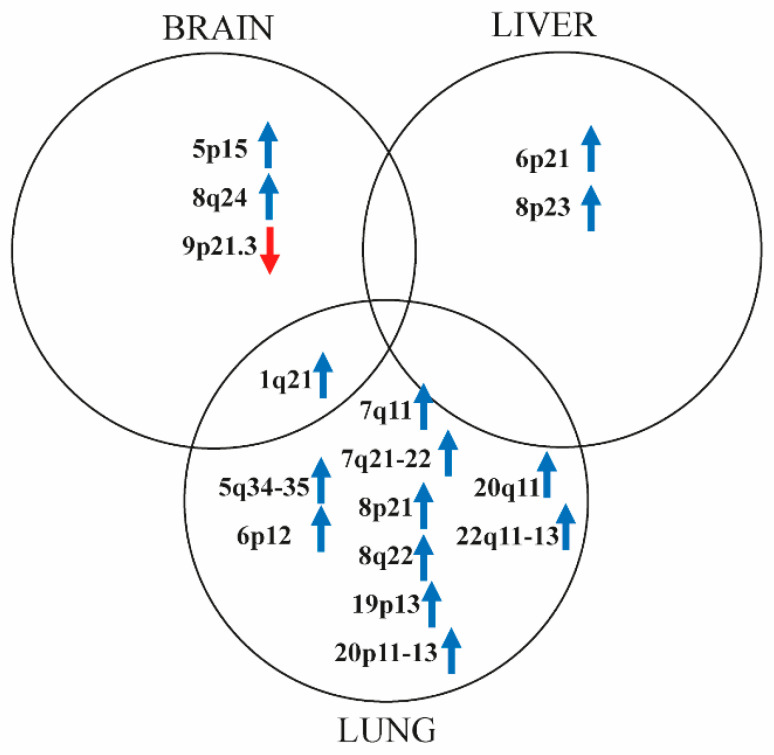
Shared and specific chromosomal regions distorted by high-copy number gains (CN ≥ 4) and homozygous losses. Blue and red arrows represent copy number gains and number losses, respectively.

**Figure 5 cancers-13-05984-f005:**
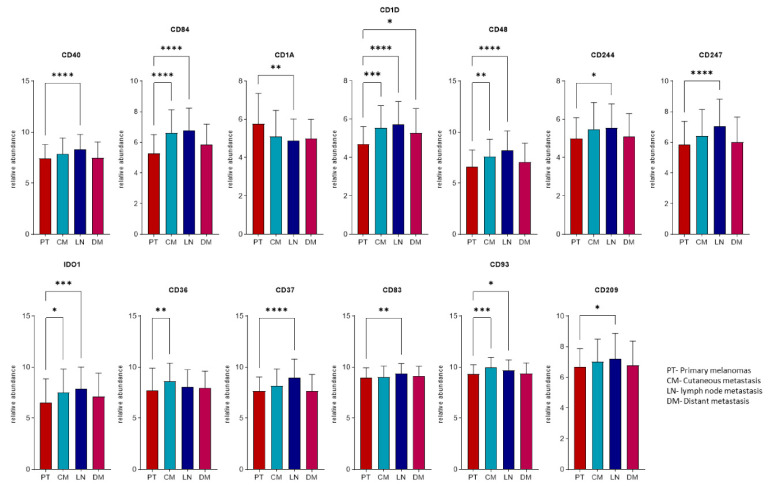
Relative abundance levels of the transcripts differentially expressed in at least one group of metastases relative to primary melanomas. Asterix represents the level of significance (* *p* < 0.05; ** *p* < 0.01; *** *p* < 0.001; **** *p* < 0.0001) using analysis of variance (ANOVA) test adjusted with the Benjamini–Hochberg method and an additional Tukey–Kramer post hoc test for each ANOVA analysis.

**Figure 6 cancers-13-05984-f006:**
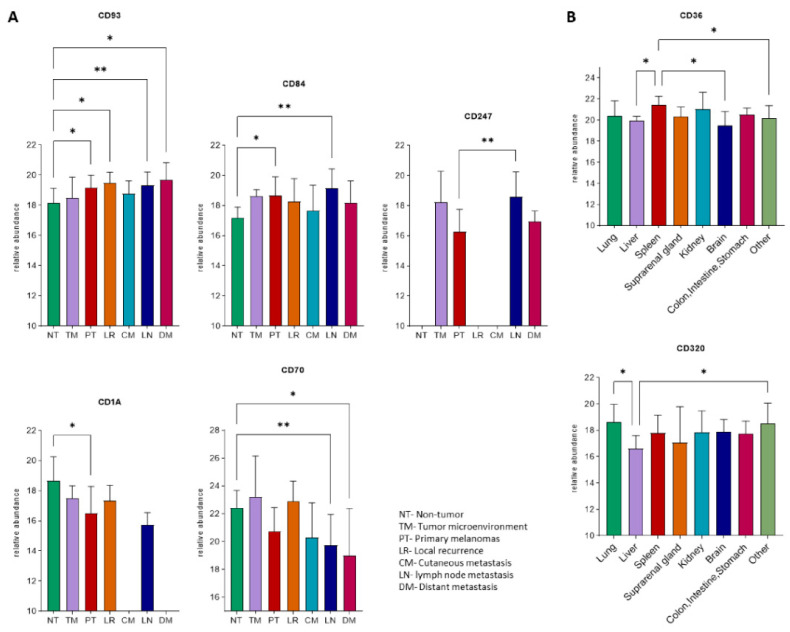
Relative abundance levels of proteins with significant differences between sample origin-based groups in the proteomics cohorts. (**A**) Proteins with significant differences between at least two groups from the prospective cohort. (**B**) Proteins with significant differences between at least two groups from the postmortem cohort. Asterisks represent the level of significance (* *p* < 0.05; ** *p* < 0.01) using Analysis of Variance (ANOVA) test adjusted with the Benjamini–Hochberg method and an additional Tukey–Kramer post hoc test for each ANOVA analysis.

**Table 1 cancers-13-05984-t001:** Metastatic melanoma patient cohort.

Primary Tumor	*N* = 10 (100%)
**Breslow thickness**: range+SEM	4.44 (1.0–9.25)
<1.0	1 (10)
1.1–2.0	2 (20)
2.1–4.0	2 (20)
>4.1	5 (50)
**Histological types**	
SSM	3 (30)
NM	2 (20)
LMM	1 (10)
unclassified	4 (40)
**Anatomical location**	
trunk	2 (20)
head and neck	2 (20)
extremities	6 (60)
**Mutational status**	
BRAFV600K/E	6 (60)
NRASQ61L/R	3 (30)
BRAF/NRAS/KIT wild type	1 (10)
**Gender**	
male	6 (60)
female	4 (40)
**Age (years)**	47.6 + 18.1
**Overall survival (months)**: range + SD	69.3 + 39.6
**Treatment**	
IFN2α	7 (70)
BRAFi	3 (30)
DTIC	2 (20)
cisplatinum	2 (20)
**Metastasis**	***N* = 28 (100%)**
brain	10 (36)
liver	9 (32)
lung	9 (32)

Abbreviations: BRAFi, BRAF inhibitor; DTIC, Dacarbazine; LMM, lentiginous melanoma; NM, nodular melanoma; SSM, superficially spreading melanoma.

**Table 2 cancers-13-05984-t002:** Copy number changes of historical genes associated with the metastatic ability of malignant melanoma.

Gene	Locus	BRAIN			LIVER			LUNG		
		Mean CN	Type of Change	Frequency (%)	Mean CN	Type of Change	Frequency (%)	Mean CN	Type of Change	Frequency (%)
CDKN2A	9p21.3	0	hoCNL	20	-	-	-	0	hoCNL	22
CDKN2B		0	hoCNL	10	-	-	-	0	hoCNL	11
TERT	5p15.33	4.5	amp.	20	-	-	-	-	-	-
MAPK15	8q24.3	5	amp.	10	4.5	amp.	22	4.66	amp.	33
TWIST1	7p21.1	5	amp.	10	5	amp.	11	4.75	amp.	44
TWIST2	2q37.3	-	-	-	-	-	-	4	amp.	11
SNAI1	20q13.13	-	-	-	-	-	-	4	amp.	11
SNAI2		-	-	-	5	amp.	22	4.66	amp.	22
S100A9	1q21.3	-	-	-	6	amp.	11	4.66	amp.	33
S100A10		-	-	-	5	amp.	11	4.66	amp.	33
S100A11		-	-	-	5	amp.	11	4.66	amp.	33
S100A12		-	-	-	6	amp.	11	4.66	amp.	33
NEDD9	6p24.2	-	-	-	-	-	-	4	amp.	22
TEAD1	11p15.3	-	-	-	-	-	-	4	amp.	11
TEAD2	19q13.33	-	-	-	-	-	-	4	amp.	22
TEAD4	12p13.33	-	-	-	-	-	-	4	amp.	22
CDK6	7q21.2	-	-	-	-	-	-	4	amp.	44
MAPK1	22q11.22	-	-	-	-	-	-	4	amp.	22
ABL2	1q25.2	-	-	-	-	-	-	4.66	amp.	33
HGF	7q21.11	6	amp.	10				8.33	amp.	33
MET	7q31				6	amp.	11	9	amp.	11

The amplification of TERT was specific for brain metastases. MAPK15 and TWIST1 amplification was observed in all metastatic sites; brain and lung metastases represented the homozygous loss of cell cycle regulators CDKN2A/B. SNAI2 and the S100A gene amplification had been called in liver and lung metastases. HGF amplification was detected in brain and liver metastases, while MET amplification were found in all metastases, except for the brain. The rest of the represented alterations were specific only to lung metastases. Abbreviations: CN, copy number; hoCNL, homozygous loss; amp., amplification.

**Table 3 cancers-13-05984-t003:** Chromosomal regions including immune cell markers affected by amplification in melanoma metastases.

Gene	Locus	BRAIN			LIVER			LUNG		
		Mean CN	Type of Change	Frequency (%)	Mean CN	Type of Change	Frequency (%)	Mean CN	Type of Change	Frequency (%)
CD160	1q21.1	4	amp.	10	-	-	-	-	-	-
CD40	20q13.12	-	-	-	5.5	amp.	22	6	amp.	22
CD83	6p23	-	-	-	5	amp.	11	4	amp.	22
CD172	20p13	-	-	-	5	amp.	11	4.5	amp.	22
CD1A-E	1q23.1	-	-	-	-	-	-	4.66	amp.	33
CD48	1q23.3	-	-	-	-	-	-	4.66	amp.	33
CD84		-	-	-	-	-	-	4.66	amp.	33
CD244		-	-	-	-	-	-	4.66	amp.	33
CD247	1q24.2	-	-	-	-	-	-	4.66	amp.	33
IDO1	8p11.21	-	-	-	-	-	-	5	amp.	22
IDO2		-	-	-	-	-	-	5	amp.	22
BCR	22q11.23	-	-	-	-	-	-	4	amp.	33
IL17R	22q11.1	-	-	-	-	-	-	4	amp.	22
CD36	7q21.11	-	-	-	-	-	-	4.25	amp.	44
CD37	19q13.33	-	-	-	-	-	-	4	amp.	22
CD70	19p13.3	-	-	-	-	-	-	4	amp.	11
CD93	20p11.21	-	-	-	-	-	-	4.5	amp.	22
CD209	19p13.2	-	-	-	-	-	-	4	amp.	11
CD320		-	-	-	-	-	-	5	amp.	11

CD160 amplification was characteristic in only the brain, while CD40, CD83, CD172 amplifications were a common phenomenon between liver and lung metastases. Abbreviation: CN, copy number; amp., amplification.

**Table 4 cancers-13-05984-t004:** Comparison of copy number changes and mutant allele frequency in matched primary tumors and metastasis pairs.

Scheme.	Site	Driver Mutation in DM	% of Mutant Cells * (P **)	% of Mutant Cells * (DM ***)	DM/P Ratio	Alteration of Driver Mutant Gene in Primary Tumor	Alteration of Driver Mutant Gene in Metastasis
BRAF primary							
17	liver	BRAF	29.1	69.1	2.37	n.d.	CN 4 + LOH
	lung	BRAF		77.9	2.68		CN 3 + LOH
19	liver	BRAF	40.8	50.4	1.24	CN 3	CN 3
	lung	BRAF		55.3	1.36		CN 3
24	brain	BRAF	38.4	28.6	0.74	no change	no change
	brain	BRAF		24.0	0.63		no change
28	brain	BRAF	21.2	35.7	1.68	n.d.	CN 3
	brain	BRAF		32.8	1.55		no change
	lung	BRAF		37.5	1.77		CN 3
31	liver	WT	27.4	8.3	0.30	n.d.	no change
	lung	BRAF		29.5	1.08		no change
32	liver	BRAF	4.0	14.9	3.73	no change	SNP
	lung	BRAF		33.1	8.28		CN 5
36	brain	BRAF	23.7	68.6	2.89	n.d.	CN 3
48	brain	BRAF	40.3	50.9	1.26	no change	no change
54	lung	BRAF	2.2	80.0	36.36	n.d.	CN 3 + LOH
56	liver	BRAF	4.7	14.0	2.98	CN 3	CN 3
	lung	BRAF		54.1	11.51		CN 4
57	brain	BRAF	11.0	24.8	2.25	no change	CN 3
	liver	BRAF		28.0	2.55		CN 3
	lung	BRAF		37.8	3.44		CN 3
59	lung	BRAF	4.3	75.4	17.53	no change	CN 5
NRAS primary							
20	liver	NRAS	4.6	29.2	6.35	−	
53	brain	NRAS	15.8	49.4	3.13	no change	LOH
55	brain	NRAS	35.4	22.8	0.64	LOH	no change
	liver	NRAS		38.0	1.07		no change
WT primary ****							
33	brain	WT	0.0	0.0	−	n.r.	n.r.

Asterix represents: * corrected to tumor cell content of the sample, ** primary tumor, *** distant metastasis, **** no minor cell population with BRAF or NRAS mutation was found in both primary and metastatic samples. Abbreviations: n.d., no data available; n.r., not relevant.

**Table 5 cancers-13-05984-t005:** Correlation between *BRAF* gene copy number alteration and mutant allele frequency in *BRAF* mutant metastases.

*BRAF*	Copy Number State (N)			
	CN2 (7)	CN3 (11)	CN4 (2)	CN5 (2)
MAF * (mean ± SE)	27.00 ± 5.16	46.36 ± 6.63	61.60 ± 7.50	52.25 ± 21.25
heterozygous	1	2	1	0
homozygous	0	3	1	1
subclonal	6	6	0	1
Correlation coefficient ** (*p*-value)	0.540 (0.009)			

Asterisks represent: * mutant allele frequency, ** Spearman’s Rho correlation, two-tailed.

## Data Availability

The data presented in this study are accessible through GEO Series accession number GSE185165 (https://www.ncbi.nlm.nih.gov/geo/query/acc.cgi?acc=GSE185165, accessed on 30 September 2021).

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
