# Peer review of "Organ Specific Copy Number Variations in Visceral Metastases of Human Melanoma"

_cancers, 2021, doi:10.3390/cancers13235984_

Round 1

Reviewer 1 Report

An array-based copy number variations analysis was performed on autopsy-based visceral metastasis biobank samples, particularly considering brain, lung, and liver. This study was completed by proteomic analysis. Brain and lung metastases were characterized by the presence of HGF/MET autocrine loop. Manuscript is of particular interest and sheds new light on melanoma diagnosis and possible treatment. The study is well conducted and the only criticism can be represented by the patient cohort made either by male and female subjects. It is well known that gender disparity characterized melanoma development and progression, thus a revision of the results considering sex of the patients might be very important.

Reviewer 2 Report

Dear Authors,

Thank you for submitting a manuscript titled “Organ specific copy number variations in visceral metastases of human melanoma”. Authors report new findings of organ specific copy number variations of melanoma using patient samples. It is well organized and written as well as very informative. However, there are some minor concerns before accepting as follows:

Minor concerns:

  1. On line 105-107, authors describe about copy number gains at chromosome 1, 5p, 9q, 14, 16p, 17q, and 22q. In Figure 1, it looks that some more copy number gains are seen such as chromosome 2, 3, 13, 15, and 18 in metastatic melanoma with compared to primary melanoma. Also, it looks that some more copy number losses are seen such as chromosome 2, 3, 4, 5, 12, 15, 17, 18 and 21 in metastatic melanoma. What do authors think?
  2. In Figure 3, numbers in each cell has two numbers. What does this mean? Duplicate of data? Some similar data are seen on Table2 and Table 3.
  3. Abbreviations are not properly explained. Some full terms of the abbreviations are explained in the later part. Abbreviated phrases should be written in full the first time that they are used.
  4. Some abbreviations such as WGS, WES, DTIC, and IGM are not explained. Please show the full terms.
  5. On line191, is the description “S1009/10/11/12, S100A10” typo?
